# Design, Synthesis and Phenotypic Profiling of Simplified Gedatolisib Analogues

**DOI:** 10.3390/ph16020209

**Published:** 2023-01-30

**Authors:** Caroline Marques Xavier Costa, Cristiane Aparecida-Silva, Luis Eduardo Reina Gamba, Thalita Neves de Melo, Gisele Barbosa, Manoel Oliveira de Moraes Junior, Victoria Regina Thomaz de Oliveira, Carolinne Souza de Amorim, João A. Moraes, Eliezer Jesus Barreiro, Lídia Moreira Lima

**Affiliations:** 1Laboratório de Avaliação e Síntese de Substâncias Bioativas (LASSBio^®^), Instituto Nacional de Ciência e Tecnologia de Fármacos e Medicamentos (INCT-INOFAR), Universidade Federal do Rio de Janeiro, CCS, Cidade Universitária, Rio de Janeiro 21941-971, RJ, Brazil; 2Programa de Pós-Graduação em Farmacologia e Química Medicinal, Instituto de Ciências Biomédicas, Universidade Federal do Rio de Janeiro, Rio de Janeiro 21941-909, RJ, Brazil; 3Laboratório de Biologia Redox (LABIO-RedOx^®^), Instituto de Ciências Biológicas, Universidade Federal do Rio de Janeiro, Rio de Janeiro 21941-902, RJ, Brazil

**Keywords:** cancer, PI3K inhibitors, gedatolisib, flow cytometry, MTT

## Abstract

Targeted antitumour therapy has revolutionized the treatment of several types of tumours. Among the validated targets, phosphatidylinositol-3 kinase (PI3K) deserves to be highlighted. Several PI3K inhibitors have been developed for the treatment of cancer, including gedatolisib (**4**). This inhibitor was elected as a prototype and molecular modifications were planned to design a new series of simplified gedatolisib analogues (**5a-f**). The analogues were synthesised, and the comparative cytotoxic activity profile was studied in phenotypic models employing solid and nonadherent tumour cell lines. Compound **5f** (LASSBio-2252) stood out as the most promising of the series, showing good aqueous solubility (42.38 μM (pH = 7.4); 39.33 μM (pH = 5.8)), good partition coefficient (cLogP = 2.96), cytotoxic activity on human leukemia cell lines (CCRF-CEM, K562 and MOLT-4) and an excellent metabolic stability profile in rat liver microsomes (t_1/2_ = 462 min; Clapp = 0.058 mL/min/g). The ability of **5f** to exert its cytotoxic effect through modulation of the PI3K pathway was demonstrated by flow cytometry analysis in a comparative manner to gedatolisib.

## 1. Introduction

Phosphatidylinositol-3 kinases (PI3Ks) are intracellular lipid kinases that catalyse phosphorylation of phosphatidylinositol-4,5-biphosphate (PI(4,5)P2) to phophatidylinositol-3,4,5-triphosphate (PI(3,4,5)P3), a second messenger that activates serine/threonine kinase AKT, regulating numerous cellular functions including cell cycle, apoptosis and glucose metabolism. Based mainly on their structure and substrate preference, PI3Ks are divided into classes I, II and III. They are activated by receptor tyrosine kinases and G-protein-coupled receptors. Class I is the most studied and consists of a regulatory subunit in complex with a p110 catalytic subunit (p110α, p110β, p110γ, p110δ) [1,2,3,4].

In human cancer, the PI3K pathway (PI3K-AKT-mTOR) is most activated by PI3Kα mutation, overexpression of growth factor receptors with tyrosine kinase activity and PTEN gene inactivation. Given the key role of PI3K in cancer, it is considered a promising target for cancer therapy, and significant efforts have been made to develop antitumour PI3K inhibitors. Those inhibitors can be classified as pan inhibitors (copanlisib, **1**), selective PI3K inhibitors (idelalisib and duvelisib, **2** and **3**) and dual PI3K/mTOR inhibitors (gedatolisib, **4**) (Figure 1) [5,6,7]. Gedatolisib (**4**, PKI-587) is a bis(morpholino- 1,3,5-triazine) pan dual inhibitor of PI3K and mTOR developed by Pfizer. It has displayed potent cytotoxic activity against MDA-361 breast adenocarcinoma and PC-3 prostate cancer lines, with potency of 10 nM and 13 nM, respectively. In enzymatic assays, gedatolisib inhibits class I PI3K isoforms α, β, δ and γ with an IC_50_ of 0.4 nM, 6 nM, 6 nM and 5.4 nM, respectively. It also inhibits mTOR activity with an IC_50_ of 1.6 nM [8,9,10].

In this work, a series of simplified gedatolisib analogues (**5a-f**) was designed, synthesised and evaluated to establish their comparative cytotoxic activity and their ability to modulate the PI3K pathway in phenotypic models using human tumour cell lines.

The design concept of compounds **5a-f** (Figure 2) considered the maintenance of the morpholino-1,3,5-triazine scaffold, the simplification of the functionalized phenyl urea subunit by a phenylpiperazine moiety, and the isosteric replacement of the oxygen atom of the morpholine ring by sulphur (S, **5a**), methylene (CH_2_, **5b**) and NH (**5c**) (Figure 2) [11,12]. Further, homologation strategies [13] were adopted to design compounds **5d**, **5e** and **5f**.

## 2. Results and Discussion

### 2.1. Chemistry

The simplified gedatolisib analogues (**5a-f**) were synthesised through a three-step linear synthesis, exploring cyanuric chloride (**6**) as starting material. As depicted in Figure 3, the first step was based on an aromatic nucleophilic mono-substitution using morpholine as a nucleophile. A second aromatic nucleophilic substitution (SNAr) was performed using phenylpiperazine to obtain the key intermediate **8**. The target compounds **5a-f** were obtained in overall yields ranging from 40 to 48% from the SNAr reaction between intermediate **8** and the preselected amines (Figure 3).

All compounds were characterized by IR, ^1^H-NMR, ^13^C-NMR and HR-MS. The relative purity was determined by HPLC.

Once synthesised and characterized, compounds had their aqueous solubility determined at pH 5.8 (duodenum) and 7.4 (plasma). The results are shown in Table 1 in a comparative manner to the cLogP, cLogD_5.8_ and cLogD_7.4_ values predicted in silico using ACD/Percepta^®^ software (ACD/LABS, Toronto, Ontario). For comparison purposes, the compound **9** containing two morpholine rings (described in the literature under the code CAS-942789959) was added to the studies. The isosteric exchange of oxygen (**9**) for sulphur (**5a**) and methylene (**5b**) results in increased partition coefficient and decreased aqueous solubility, whereas the exchange by the NH group (**5c**) results in a significant increase in solubility and similar cLogP. The homologation of compound **5c** by the introduction of a methyl group (**5d**) or hydroxyethyl group (**5e**) or by a methylene to the piperazine ring (**5f**) results in a small decrease in aqueous solubility and a respective increase in cLogP values. However, all three compounds (**5d-f**) showed solubility greater than 30 μM at both duodenum and plasma pH values (Table 1).

### 2.2. Pharmacological Experiments

#### 2.2.1. Cell Viability by MTT and CC_50_ Determination

To determine the cytotoxic effect of the new simplified gedatolisib analogues (**5a-f**), the MTT cell viability assay was performed using the following human cell lines: PC-3, MCF7, K562, CCRF-CEM and MOLT-4. In a first approach, the assay was validated with two standard drugs: gedatolisib and GSK-1059615.

In the CCRF-CEM (human leukemia cell line), gedatolisib (**4**) showed a CC_50_ of 30 nM similar to the value of 23 nM described previously by Gazi and coworkers [14]. For the MCF7 (breast cancer cells), compound **4** displayed a CC_50_ of 70 nM, a value close to that found by Mallon and coworkers [15], who reported CC_50_ = 36 nM (Table 2). On the other hand, we demonstrated for the first time the cytotoxic effect of gedatolisib against PC3 (human prostate cancer cell line), MOLT-4 (T lymphoblast cell line) and K562 (myelogenous leukemia cell line). As demonstrated in Table 2, gedatolisib was inactive on K562 (CC_50_ > 24.000 nM) and showed reduced cytotoxic potency on PC3 cells (CC_50_ = 540 nM). However, it was quite potent on MOLT-4 cells with CC_50_ = 20 nM.

Although gedatolisib and GSK-1059615 have similar potency against inhibition of PI3K and mTOR isoforms [10], in the phenotypic assays GSK-1059615 proved to be less potent than gedatolisib, except on K562 cells. However, as expected, both compounds had the greatest cytotoxic effect on cell lines with mutations in the P3IK pathway (i.e., MCF7, CCRF-CEM and MOLT-4) (Table 2).

After understanding the response profile of the selected cell lines to the treatment with gedatolisib and GSK-1059615, we investigated the cytotoxic activity of compounds **5a-f** at a single concentration (30 μM). Compounds that showed a reduction in cell viability ≥ 50% were selected for CC_50_ determination. During the experiments, compounds **5a** (LASSBio-2250), **5b** (LASSBio-2251) and **9** (CAS-942789959) were insoluble and precipitated on the culture medium. Therefore, they were evaluated also at the screening concentration of 1 μM. All three compounds were inactive at 1 μM. On the other hand, the analogues **5c** (LASSBio-2247), **5d** (LASSBio-2248) and **5f** (LASSBio-2252) had a cytotoxic effect on leukemia cell lines and were selected for determination of their 50% cytotoxic concentration (CC_50_) (Table 3).

Considering that compounds **5a-f** were designed as simplified gedatolisib analogues, the absence of a cytotoxic effect on solid tumour lines, particularly on MCF-7, was not expected. MCF-7 strains are mutated with overexpression of the PI3KCA gene, so they are strains sensitive to PI3Kα inhibitors (p110α). As previously informed, gedatolisib, which inhibits PI3Kα with IC_50_ = 0.4 nM [10], showed a CC_50_ on MCF-7 cells of 70 nM (Table 2). This result suggests that compounds **5a-f** do not modulate PI3Kα isoform activity or do so with low potency. Considering this hypothesis, cytotoxic response concentration studies were maintained with MCF-7 and PC-3 cells.

In the concentration response study, using the 72 h MTT cell viability assay, **5c** and **5d** showed the maximum effect (E_max_) on PC3 cells of 58% and 72%, respectively, and displayed CC_50_ = 92.29 and 91.31 μM, respectively. Similar results were found on MCF-7 (E_max_ = 57% and 77%, respectively, and CC_50_ > 100 μM) (Table 3). These results support the hypothesis that such compounds are weak inhibitors of PI3Kα.

Compound **5f** (LASSBio-2252), bearing a homopiperazine instead of a piperazine subunit, showed a more favourable cytotoxic activity profile. It displayed CC_50_ of 62.15 μM and 37.04 μM against PC-3 e MCF-7 cell lines, respectively (Table 3). Similarly, LASSBio-2252 (**5f**) was the one with the highest cytotoxic potency on K562, CCRF-CEM and MOLT-4 strains, with CC_50_ values of 23.41 µM, 6.25 µM and 9.76 µM, respectively (Table 3).

Trying to establish a suitable experimental condition for the flow cytometry study, LASSBio-2252 (**5f**) and gedatolisib (**4**) were tested with the 48 h MTT assay on the CCRF-CEM and MOLT-4 cell lines. As depicted in Table 4, there is approximately a 10-fold loss of cytotoxic potency of gedatolisib when assayed at 48 h, whereas the loss of potency of compound **5f** was less than 4-fold for MOLT-4 and less than 2-fold for CCRF-CEM (Table 4 versus Table 3). However, there is a clear impact on reducing the maximum cytotoxic effect (E_max_) induced by **5f** on CCRM-CEM cells.

#### 2.2.2. Evaluation of the Cytotoxic Selectivity Index

The cytotoxic selectivity index (IS) represents how selective the compound is for tumour cells in relation to nontumour cells, indicating less possibility of adverse effects [16]. The IS of gedatolisib and compound **5f** (LASSBio-2252) was calculated by the ratio between the CC_50_ on tumour cells and the CC_50_ on nontumour cells. The latter was determined by the 72 h MTT assay using human peripheral blood mononuclear cells (PBMCs). At the highest concentration studied (i.e., 50 μM), gedatolisib was not cytotoxic to PBMCs, reducing the viability of normal mononuclear cells by only 33% (E_max_ = 33%). Concentrations higher than 50 μM could not be studied due to the insolubility of gedatolisib. A similar result was observed for LASSBio-2252 (**5f**). It was assayed at five concentrations, ranging from 0.03 to 100 µM. Even at 100 μM, LASSBio-2252 (**5f**) was unable to induce a cytotoxic effect above 26% (E_max_ = 26%) on PBMCs (Appendix A).

#### 2.2.3. Analysis of Cell Viability by Flow Cytometry at 48 and 72 h

To confirm the cytotoxic effect in a nonmetabolic assay, the comparative cytotoxic potency of **5f** (LASSBio-2252) and gedatolisib (**4**) was restudied, using the Guava Muse Luminex^®^ benchtop cytometer cell viability kit. The assay was conducted using different concentrations of **4** and **5f** (0.003 to 100 µM) and MOLT-4, CCRF-CEM and MCF7 cell lines.

The cytotoxic potency of gedatolisib (**4**) in the Muse^®^ Count & Viability Kit at 72 h was lower than in the 72 h MTT (Table 5). In fact, gedatolisib (**4**) showed 100-fold lower potency than that determined by the MTT method on CCRF-CEM (CC_50_ = 6.15 µM) and MOLT-4 (CC_50_ = 1.15 µM). For the MCF7 cells, the loss of potency was so great that the CC_50_ could not be determined. At 50 μM, gedatolisib displayed a maximum effect (E_max_) of only 54%. There was also a loss of potency of LASSBio-2252 (**5f**), but the difference did not exceed 10-fold for both CCRF-CEM and MOLT-4. Compound **5f** was inactive on MCF-7 (Table 5).

These data confirm the cytotoxic effect of the simplified gedatolisib analogue **5f** against PI3K mutated cells lines, using a different methodology than MTT, and whose observed cytotoxic effect depends on the disruption of the cell membrane and no longer on the metabolic activity of mitochondria, on which MTT is based.

The Muse^®^ Count & Viability Kit was also used to analyse cell viability by flow cytometry at 48 h (Table 5). No significant differences in gedatolisib potency were identified on CCRF-CEM and MOLT-4 when compared to the assay performed at 72 h (Table 5). The data suggest that the cytotoxicity assessed by the MTT method is incubation time dependent (48 h or 72 h, Table 2, Table 3 and Table 4), whereas the cytotoxicity measured by the viability and cell counting kit is not incubation time dependent (Table 5). Similar results were found for compound **5f**. Another intriguing fact is that the difference in cytotoxic potency between gedatolisib and **5f** was about 10-fold by this method, whereas it was 100-fold to 700-fold by 48 h and 72 h via the MTT method, respectively.

#### 2.2.4. Analysis of Phosphatidylserine Externalization and Cell Membrane Integrity

For quantitative analysis of live, early and late apoptosis, and cell death on both the CCRF-CEM and MOLT-4 cell lines, treated with gedatolisib and compound **5f** (LASSBio-2252), the Muse^®^ Annexin V and Dead Cell Kit was used. Compounds were studied using three concentrations. The first was based on the average of the CC_50_ values obtained in 48 h and 72 h (Table 5), a second concentration was 10 times lower, and a third was 10 times higher.

As can be seen in Figure 4, at their CC_50_ values both gedatolisib and its simplified analogue **5f** reduced cell viability by about 50% and caused about 50% total apoptosis of the CCRF-CEM. In contrast, on the MOLT-4 compound **5f** reduced the number of viable cells by only 14% when tested at its CC_50_ value and by 80% when tested at 10-fold concentration (Figure 5). On the other hand, gedatolisib reduced cell viability by 90% already at a concentration 10 times lower than its CC_50_. However, both compounds led to late apoptosis of MOLT-4 cells with increasing concentrations. These results suggest that, despite the difference in potency, gedatolisib and its analogue **5f** (LASSBio-2252) displayed similarity in the cellular mechanism of action against MOLT-4 and CCRF-CEM.

#### 2.2.5. Cell Cycle Assessment

To understand in which phase of the cell cycle the compounds can act, the Muse^®^ Cell Cycle Kit was used to allow the quantitative measurement of the percentage of cells in the G0/G1, S and G2/M phases of the cell cycle. The assay was performed using the CCRF-CEM cell line treated with doxorubicin, vincristine, gedatolisib and LASSBio-2252 (**5f**). As demonstrated in Figure 6, doxorubicin at the concentration of 0.5 µM behaves as expected, promoting nonspecific action on the cell cycle. Vincristine at 5 µM promotes increased arrest in the G2/M phase, due to its action as an antimicrotubule agent. The concentration choice was based on previous data in the literature [17,18,19,20]. Gedatolisib led to cell cycle arrest in the G0/G1 phase at both concentrations tested (0.6 µM and 6 µM), in agreement with the behaviour expected for PI3K inhibitors [21]. Similarly, LASSBio-2252 (**5f**) led to cell cycle arrest in G0/G1 when tested using its CC_50_ value (i.e., 50 µM).

#### 2.2.6. Analysis of PI3K Activation and Phosphorylation

To assess whether the cytotoxic effect is due to the modulation of the PI3K pathway, the evaluation of the activity on the PI3K pathway was investigated by using the Muse^®^ PI3K Activation Dual Detection Kit on the CCRF-CEM and MOLT-4 cell lines. As expected, the treatment of CCRF-CEM cells with gedatolisib, at a concentration equivalent to its CC_50_ (~6 μM) and 10 times lower than its CC_50_ (0.6 μM) (Table 4), increased the percentage of cells in the inactivated state (i.e., unable to phosphorylate the AKT substrate) by about 75% (Figure 7), whereas LASSBio-2252 (**5f**) raised the percentage of inactivated cells to 30% and 65% when tested at its CC_50_ and at a concentration 10 times higher than its CC_50_, respectively (Figure 7). Interestingly, in MOLT-4 cells, even at the concentration 10 times lower than its CC_50_, LASSBio-2252 (**5f**) reduced PI3K-mediated phosphorylation by 79% (Figure 8). 

#### 2.2.7. Microsomal Metabolic Stability

To investigate the metabolic stability of LASSBio-2252 (**5f**), the compound was incubated with rat liver microsomes in the presence and in the absence of the NADPH-generating system. Metabolization rate, half-life (t_1/2_) and apparent intrinsic clearance (Clapp) were calculated. As shown in Table 4, this simplified analogue of gedatolisib exhibits high metabolic stability, with t_1/2_ superior to 6 h and low clearance rate (Table 6).

## 3. Material and Methods

### 3.1. Synthesis and Characterization

All reagents and solvents were purchased from commercial suppliers. The reactions were monitored by thin-layer chromatography, which was performed on aluminium sheets precoated with silica gel 60 (HF-254, Merck) to a thickness of 0.25 mm. The chromatograms were viewed under ultraviolet light (254–365 nm). For column chromatography, Merck silica gel (230–400 mesh) was used. ^1^H-NMR, ^13^C-NMR and DEPT-135 spectra were determined in deuterated dimethyl sulfoxide using a Varian NMR at 400 MHz and 500 MHz. Chemical shifts are given in parts per million (d) with tetramethylsilane as the internal standard, and coupling constant values (J) are given in Hertz (Hz). Signal multiplicities are represented by: s (singlet), d (doublet), t (triplet), q (quadruplet), m (multiplet) and br (broad signal).

Infrared (IR) spectra were obtained with a FTIR Thermo Scientific™ Nicolet™ iS10 spectrophotometer in ATR mode using ruby crystal support, and the absorption values were expressed in inverse centimetres (cm^−1^). Melting points were determined by differential scanning calorimetry with a Shimadzu DSC-60 apparatus until 300 °C (heating rate: 20 °C/min). DSC-60 equipment calibration was performed using indium (In) as the standard (m.p. 157.2 °C).

The purity of compounds was determined by HPLC (95%) using the Shimadzu—LC20AD apparatus, a Kromasil 100–5C18 (4,6 mm x 6250 mm) column and the SPD-M20A detector (diode array) at 254 nm for quantification of analyte in a 1 mL/min constant flux. The injector was programmed to inject a volume of 20 µL. Mass spectrometry was obtained by positive ionisation on a BrukerAmaZon SL, and data were analysed in Compass 1.3.SR2 software.

#### 3.1.1. Procedure of the Preparation of 4-(4,6-Dichloro-1,3,5-triazin-2-yl)morpholine (**7**) (Adapted from Reference [22])

Cyanuric chloride (**6**) (0.5g, 2.71 mmol) was solubilized in 15 mL of THF and subjected to stirring under an ice bath. A mixture of morpholine (0.2 mL, 2.43 mmol) and TEA (0.4 mL, 2.43 mmol) dissolved in 15 mL of THF was added dropwise with the aid of a standard addition funnel. After the addition, the reaction was kept under constant stirring at 0 °C for 6 h. After the end of the reaction, the reaction mixture was concentrated to half the volume under reduced pressure, and crushed ice was added, leading to the precipitation of a white solid that was filtered and washed with 10 mL of cold distilled water. The product obtained was considered pure and was used in the next step without purification. Compound **7** was obtained as white powder in 91% yield and m.p. 139–140 °C. The data agree with the literature [23]. 1H-NMR (500 MHz, CDCl3) δ (ppm): 3.876 (4H, t, J = 5) e 3.742 (4H, t, J = 5). 13C-NMR (500 MHz, CDCl3) (δ, ppm): 170.56; 164.23; 66.50; 44.59.

#### 3.1.2. Procedure of the Preparation of 4-(4-Chloro-6-(4-Phenylpiperazin-1-yl)-1,3,5-triazin-2-yl)morpholine (**8**) (Adapted from Reference [22])

Compound **7** (300 mg; 1.28 mmol) was solubilized in THF (15 mL) and subjected to stirring under an ice bath. A mixture of phenylpiperazine (1.1 equivalents) and TEA (1.1 equivalents) solubilized in THF (15 mL) was added dropwise with the aid of a standard addition funnel. After the addition, the reaction was kept under constant stirring at 0 °C. After 24 h, the mixture was concentrated, and the residue was treated with 5% Na_2_CO_3_ until pH 7–8. The product was extracted with CHCl_3_. The organic phase was dried with Na_2_SO_4_ and concentrated to a pale-yellow oil, which was filtered over SiO_2_ to obtain 414 mg of the translucent oil as a product. Compound **8** was obtained in 90% yields.

1H-NMR (400 MHz, CDCl3) δ (ppm): 7.314 (2H, t, J = 7.5); 7.053 (2H, sl); 6.974 (1H, m); 4.017 (4H, sl); 3.232 (4H, m) 3.800 (4H, m); 3.715 (4H, m); 3.232 (4H, m). 13C-NMR (500 MHz, CDCl3) (δ, ppm): 169.89; 164.67; 164.53; 129.52; 117.74; 66.69; 49.90; 44.01; 43.15.

#### 3.1.3. General Procedure for the Preparation of N-Substituted Triazines (**5a-f**) (Adapted from Reference [22])

The key intermediate **8** (300 mg; 0.83 mmol) was dissolved in THF (10 mL). A solution of the corresponding amine (1.1 eq) and TEA (1.1 eq) dissolved in THF (5 mL) was added. After the addition, the reaction was kept under constant stirring at reflux for 18–24 h. After the end of the reaction, the mixture was concentrated, and the residue was treated with 5% Na_2_CO_3_ until pH 7–8. The product was extracted with CHCl_3_. The organic phase was dried using Na_2_SO_4_ and purified by column chromatography to obtain compounds **5a-f** as white solids. Yields and characterization pattern are described below:

##### 4-(4-(4-Phenylpiperazin-1-yl)-6-(piperidin-1-yl)-1,3,5-triazin-2-yl)morpholine (LASSBio-2247, **5a**)

Compound **5a** was obtained in 50% yield as a white powder. Melting point: 177 °C.

^1^H-NMR (400 MHz, CDCl_3_) δ (ppm): 7.27 (H-27, H-29, t, J = 6.6 Hz, 2H), 6.95 (H-26, H-30, d, J = 8.0 Hz, 2H), 6.88 (H-28, t, J = 7.5 Hz, 1H), 3.93–3.90 (H-15, H-19, H-20, H-24, m, 8H), 3.75 (H-11, H-13, m, 4H), 3.71–3.70 (H-10, H-14, m, 4H), 3.18 (H-16, H-18, t, J = 5.0 Hz, 4H), 3.04 (H-21, H-23, m, 4H).

^13^C-NMR (100 MHz, CDCl_3_) δ (ppm): 165.49 (C-4), 165.36 (C-2), 165.33 (C-6), 151.50 (C-25), 129.31 (C-27, C-29), 120.26 (C-28), 116.65 (C-26, C-30), 66.96 (C-11, C-13), 49.54 (C-16, C-18), 44.14 (C-21, C-23), 43.76 (C-10, C-14), 43.20 (C-15, C-19), 41.91 (C-20,C-24).

IR (ATR-FTIR, cm^−1^): 3325 (N-H), 3065–2735 (CH2), 1230 (C-N)

95.4% purity in HPLC (R.T. = 9.48 min; MeOH/H_2_O (6:4 *v/v*), pH = 3)

MS: m/z = HR-MS (ESI) [M + H] + : 411.26196 (cal. 411.26208).

##### 4-(4-(4-Methylpiperazin-1-yl)-6-(4-phenylpiperazin-1-yl)-1,3,5-triazin-2-yl)morpholine (LASSBio-2248, **5b**)

Compound **5b** was obtained in 52% yield as a white powder. Melting point: 135 °C.

^1^H-NMR (400 MHz, DMSO-d6) δ (ppm): 7.28 (H-27, H-29, t, J = 6.6 Hz, 2H), 6.96–6,94 (H-26, H-30, d, J = 7.5 Hz, 2H), 6.88 (H-28, t, J = 7.3 Hz, 1H), 3.92–3.87 (H-15, H-19, H-20, H-24, m, 8H), 3.76–3.75 (H-11, H-13, m, 4H), 3.71–3.70 (H-10, H-14, m, 4H), 3.18 (H-16, H-18, t, J = 5.3 Hz, 4H), 2.54 (H-21, H-23, t, J = 4.7 Hz, 4H), 2.40 (H-31, s, 3H).

^13^C-NMR (100 MHz, CDCl_3_) δ (ppm): 165.61 (C-4), 165.45 (C-2), 165.38 (C-6), 151.58 (C-25), 129.28 (C-27, C-29), 120.20 (C-28), 116.62 (C-26, C-30), 67.00 (C-11, C-13), 54.82 (C-21, C-23), 49.54 (C-16, C-18), 45.91 (C-31), 43.79 (C-10, C-14), 43.21 (C-15, C-19), 42.64 (C-20,C-24).

IR (ATR-FTIR, cm^−1^): 3096–2685 (CH_2_, CH_3_), 1228–1053 (C-N),

99.4% purity in HPLC (R.T. = 2.43 min; MeOH/H_2_O (6:4 *v/v*))

HR-MS (ESI) [M + H] + : 425.27755 (cal. 425.27773).

##### 2-(4-(4-Morpholino-6-(4-phenylpiperazin-1-yl)-1,3,5-triazin-2-yl)piperazin-1-yl)ethan-1-ol (LASSBio-2249, **5c**)

Compound **5c** was obtained in 58% yield as a white powder. Melting point: 92 °C.

^1^H-NMR (400 MHz, DMSO-d6) δ (ppm): 7.28 (H-27, H-29, t, J = 6 Hz, 2H), 6.96 (H-26, H-30, d, J = 8.0 Hz, 2H), 6.88 (H-28, t, J = 7.3 Hz, 1H), 3.91 (H-15, H-19, t, J = 5.3 Hz, 4H), 3.83 (H-20, H-24, t, J = 4.5 Hz, 4H), 3.77–3.75 (H-10, H-14, m, 4H), 3.72–3.68 (H-11, H-13, H-32, m, 6H), 3.18 (H-16, H-18, t, J = 5.0 Hz, 4H), 2.62 (H-31, t, J = 5.3 Hz, 2H), 2.57 (H-21, H-23, t, J = 5.0 Hz, 4H).

^13^C-NMR (100 MHz, CDCl3) δ (ppm): 165.64 (C-4), 165.48 (C-2), 165.38 (C-6), 151.60 (C-25), 129.30 (C-27, C-29), 120.23 (C-28), 116.65 (C-26, C-30), 67.03 (C-11, C-13), 59.80 (C-31), 57.80 (C-32), 53.04 (C-21, C-23), 49.57 (C-16, C-18), 43.81 (C-10, C-14), 43.23 (C-15, C-19), 43.02 (C-20, C-24).

IR (ATR-FTIR, cm^−1^):3419 (O-H), 2847 (CH_2_), 1234 (C-N)

99.3% purity in HPLC (R.T. = 2.42 min; MeOH/H_2_O, (9:1 *v/v*) pH = 3)

##### 4-(4-(4-Phenylpiperazin-1-yl)-6-thiomorpholino-1,3,5-triazin-2-yl)morpholine (LASSBio-2250, **5d**)

Compound **5d** was obtained in 45% yield as a white powder. Melting point: 170 °C.

^1^H-NMR (400 MHz, CDCL_3_) δ (ppm): 7.32 (H-27, H-29, t, J = 7.5 Hz, 2H), 7.12 (H-26, H-30, sl, 2H), 7.00 (H-28, sl, 1H), 4.08–4.02 (H-15, H-19, H-20, H-24, m, 8H), 3.77–3.75 (H-11, H-13, m, 4H), 3.72–3.70 (H-10, H-14, m, 4H), 3.25 (H-16, H-18, s, 4H), 2.62 (H-21, H-23, t, J = 5.0 Hz, 4H).

^13^C-NMR (100 MHz, CDCl3) δ (ppm): 164.75 (C-2, C-4, C-6), 129.55 (C-27, C-29), 121.46 (C-28), 117.46 (C-26, C-30), 66.96 (C-11, C-13), 50.68 (C-16, C-18), 46.06 (C-20, C-24), 43.95 (C-10, C-14), 42.88 (C-15, C-19), 27.24 (C-21, C-23).

IR (ATR-FTIR, cm^−1^): 3066–2838 (CH_2_), 1230 (C-N).

98.3% purity in HPLC (R.T. = 14.9 min; MeOH/H_2_O (6:4 *v/v*) pH = 3)

HR-MS (ESI) [M + H] + : 428.22326 (cal. 428.22325).

##### 4-(4-(4-Phenylpiperazin-1-yl)-6-(piperidin-1-yl)-1,3,5-triazin-2-yl)morpholine (LASSBio-2251, **5e**)

Compound **5e** was obtained in 58% yield as a white powder. Melting point: 166 °C.

^1^H-NMR (500 MHz, CDCL_3_) δ (ppm): 7.29 (H-27, H-29, t, J = 8.0 Hz, 2H), 7.02–7.01 (H-26, H-30, d, J = 7.0 Hz, 2H), 6.92 (H-28, t, J = 7.0 Hz, 1H), 3.96 (H-15, H-19, s, 4H), 3.79–3.77 (H-11, H-13, m, 4H), 3.75–3.71 (H-10, H-14, H-20, H-24, m, 8H), 3.22 (H-16, H-18, t, J = 5.0 Hz, 4H), 1.65–1.64 (H-22, m, 2H), 1.58–1.57 (H-21, H-23, m, 4H).

^13^C-NMR (126 MHz, CDCl_3_) δ (ppm): 164.61 (C-2, C-4, C-6), 150.94 (C-25), 129.37 (C-27, C-29), 120.63 (C-28), 116.92 (C-26, C-30), 67.01 (C-11, C-13), 49.93 (C-16, C-18), 44.55 (C-20, C-24), 44.05 (C-10, C-14), 43.30 (C-15, C-19), 25.93 (C-21, C-23), 25.01 (C-22).

IR (ATR-FTIR, cm^−1^): 3098–2838 (CH_2_), 1228 (C-N)

99.1% purity in HPLC (R.T. = 8.7 min; MeOH/H_2_O (9:1 *v/v*) pH = 3)

HR-MS (ESI) [M + H] + : 410.26682 (cal. 410.26683).

##### 4-(4-(1,4-Diazepan-1-yl)-6-(4-phenylpiperazin-1-yl)-1,3,5-triazin-2-yl)morpholine (LASSBio-2252, **5f**)

Compound **5f** was obtained in 55% yield as a white powder. Melting point: 176 °C.

^1^H-NMR (400 MHz, CDCL^3^) δ (ppm): 7.30–7.25 (H-28, H-30, m, 2H), 6.96 (H-27, H-31, d, J = 8 Hz, 2H), 6.88 (H-29, t, J = 8.0 Hz, 1H), 3.91 (H-15, H-19, m, 4H), 3.81 (H-20, H-25, m, 4H), 3.74 (H-10, H-14, m, 4H), 3.72–3.69 (H-11, H-13, m, 4H), 3.18 (H-16, H-18, t, J = 4.0 Hz, 4H), 3.03 (H-24, sl, 2H), 2.90 (H-22, t, J = 4.0 Hz, 2H), 1.92 (H-21, m, 2H).

^13^C-NMR (100 MHz, CDCl^3^) δ (ppm): 165.61 (C-4), 165.43 (C-2), 165.32 (C-6), 151.60 (C-26), 129.27 (C-28, C-30), 120.19 (C-29), 116.63 (C-27, C-31), 67.03 (C-11, C-13), 49.55 (C-16, C-18), 48.63 (C-25), 48.23 (C-24), 47.63 (C-22), 45.45 (C-20), 43.77 (C-10, C-14), 43.19 (C-15, C-19), 28.92 (C-21).

IR (ATR-FTIR, cm^−1^):3330 (N-H), 2847–2814 (CH_2_), 1230 (C-N)

97.8% purity in HPLC (R.T. = 3.57 min; MeOH/H_2_O (6:4 *v/v*) pH = 3)

### 3.2. pH-Dependent Aqueous Solubility

A stock solution of the target compound was prepared by dissolving 1 mg in methanol, and the wavelength of greatest absorption of each compound was obtained by scanning between wavelengths from 200 nm to 500 nm. Six dilutions were prepared from the stock solution and the absorbance reading in the UV spectrophotometer (SpectraMax M5, Molecular Devices, USA). The calibration curve with linear regression was created with the values obtained, and the solubility test was measured by correlating the concentration with the absorbance obtained by ultraviolet. Each compound was added in excess at phosphate buffer solution pH 5.8 and 7.4 to obtain a supersaturated solution that was kept under stirring for 4 h at 37 °C and then was filtered through a 0.45 µm filter for reading at the corresponding wavelengths. The solubility was determined through the equation of the straight line obtained by the linear regression of the calibration curve of the compounds [24,25].

### 3.3. Cell Culture

For the evaluation of cellular cytotoxicity in cancer cell lines, the compounds were tested at concentrations ranging from 0.00003 μM to 150 μM. Immortalized cancer cells were selected: PC-3 (PTEN-mutated prostate adenocarcinoma), MCF-7 (PI3KCA-mutated breast carcinoma), K562 (P53-mutated chronic myeloid leukemia), CCRF-CEM (acute lymphoblastic leukemia with mutation in PTEN) and MOLT-4 (acute lymphoblastic leukemia with mutation in PI3KR1/PTEN). Peripheral blood mononuclear cells (PBMCs) were used to determine the cytotoxic selectivity index (SI). All immortalized cells were cultured in RPMI 1640 medium with 10% (*v/v*) FBS, streptomycin + penicillin, at 37 °C in a humidified incubator containing 5% CO_2_.

Human cancer cell lines were acquired from the Cell Bank of Rio de Janeiro—BCRJ. PBMCs were isolated from healthy patients and followed the guidelines approved by the Ethics Committee CAAE: 38257914.7.0000.5259.

### 3.4. Cancer Cell Viability Assay

Cell viability was assessed by the MTT assay. The lines selected for this study were plated according to their respective densities, which ranged from 3 × 10^4^ to 4 × 10^4^ cells per well. Subsequently, they were treated with test compounds at different concentrations (0.00003 μM to 150 μM), gedatolisib and GSK-1059615. The plates were incubated in an oven at 5% CO_2_, at 37 °C for 48 and 72 h.

After incubation, the plates are centrifuged at 440× *g* for 10 min. Then, 110 μL of the supernatant was withdrawn and each well received 10μL of the MTT solution. The plate was again incubated in an oven at 37 °C and 5% CO_2_ for 3 to 4 h. At the end of the incubation time, 100 μL of SDS-HCL detergent was added to solubilize the formazan crystals. The plate was shaken on an automatic overnight shaker and then read at a wavelength of 595 nm on the Molecular Devices Spectramax M5 plate reader. CC_50_ values were calculated from means (95% confidence interval (CI)) using GraphPad Prism^®^, version 6 [26,27].

### 3.5. PBMC Viability Assay

Donor blood was collected and EDTA (ethylenediamine tetraacetic acid) was added to the sterile laminar flow in a 1:20 ratio. In a 15 mL falcon tube, 4.5 mL of Ficoll gradient was added, and then 9 mL of blood was added over the gradient in a 1:2 ratio. The tubes were centrifuged for 40 min at a speed of 750× *g* with an acceleration of 2 and a brake of 0. The ring of mononuclear cells formed was collected and added to a new tube.

Tubes with PBMCs were centrifuged in an incomplete RPMI medium for 10 min at 580× *g* with an acceleration of 8 and a brake of 9. The supernatant was discarded, and the cells were resuspended in 1 mL of RPMI medium supplemented with 20% PBS. PBMCs were added to 96-well plates at a concentration of 1 × 10^5^ cells per well. Test compounds were added at concentrations ranging from 0.0003 to 100 µM and incubated for 72 h in a CO_2_ incubator at 37 °C [28].

After incubation, each well received 20 µL of MTT (5 mg/mL), followed by incubation for 4 h in the CO_2_ oven and centrifugation for 12 min at 1200× *g*. The supernatant from each well was taken, and 100 µL of isopropyl alcohol was added to solubilize the formazan crystals. The absorbance of the plates was measured at 570 nm in the Flex Station equipment. CC_50_ values were calculated from means (95% confidence interval (CI)) using GraphPad Prism^®^, version 6 [16,29].

### 3.6. Flow Cytometry Analysis

The benchtop cytometer Guava Muse Luminex^®^ Cell Analyzer (Merck) was used to elucidate the mechanism of action of LASSBio-2252 in the MCF-7, CCRF-CEM and MOLT-4 cells. Cell viability, cell death pattern, cell cycle and detection of PI3K activation were evaluated, and each analysis was performed using specific protocols of the Guava Muse Cell Analyzer.

Cell lines were added in 24-well plates, and each well received 500 µL of cells in suspension with density ranging from 1 × 10^5^ to 3 × 10^5^ cells per mL, in duplicate wells. After 24 h, each well received concentrations of LASSBio-2252 or the controls gedatolisib, doxorubicin and vincristine (0.003 to 100 µM) and were reincubated in an oven with 5% CO_2_ at 37 °C for 48 h and 72 h. 

#### 3.6.1. Count and Viability Test

The Count & Viability Kit (catalog n^o^. MCH100102, LUMINEX) differentiates between viable and nonviable cells based on the permeability of the dyes present in the reagent to the cell.

In total, 20 µL of the cell solution was added to 380 µL of the reagent in 1 mL Eppendorf tubes and incubated for 5 min at room temperature in the dark. Samples from three independent experiments carried out in duplicate were read on the cytometer, followed by statistical analysis of the histograms on Prism6 to determine the CC_50_.

#### 3.6.2. Analysis of Apoptosis by Annexin V

The Annexin V and Dead Cell Kit (catalog n^o^. MCH100105, Luminex) contains Annexin V, a phospholipid marker phosphatidylserine, and the molecule 7-AAD, a DNA marker for cells with a ruptured membrane. An aliquot of 100 µL of the cell solution was added to microtubes with a density of up to 5 × 10^5^ cells/mL and homogenised with 100 µL of Annexin V reagent. After incubation for 20 min at room temperature, the samples were read on the cytometer. The data with the histograms generated contain information on the percentage of viable cells, in early apoptosis, late apoptosis and dead cells. Samples from three independent experiments were performed in duplicate, and then statistical analysis was performed using the Prism6 statistical program.

#### 3.6.3. Cell Cycle Arrest

The Guava Muse Luminex^®^ Cell Cycle Kit (catalog n^o^ MCH100106, Luminex) contains the DNA marker propidium iodide (PI) and RNAse A to avoid nonspecific labelling.

After 48 h of incubation in an incubator at 37 °C with 5% CO_2_, 1 mL of the MOLT-4 cell solution and 500 µL of the CCRF-CEM cell solution, contained in the wells, were transferred to Eppendorf tubes and centrifuged at 300× *g* for 5 min. The volumes used ensured that the cell density remained within the recommended range (1 × 10^6^ to 4 × 10^6^ cells/mL) at the end of the 48 h incubation period.

The supernatant was discarded, and the pellet formed was resuspended in 1 mL of saline PBS, homogenised, and centrifuged at 300× *g* for 5 min. The formed pellet was resuspended in residual PBS by homogenization. To fix cellular events, 1 mL of ice-cold 70% alcohol was added dropwise into the tubes and vortexed at medium speed. The samples were incubated, before staining, for at least 3 h at −20 °C.

At the end of the incubation time at −20 °C, the samples were centrifuged at 300× *g* for 5 min, and then the supernatant was discarded and resuspended in 250 µL of PBS. The samples were again centrifuged at 300× *g* for 5 min and the supernatant discarded, which constituted the washing step. The pellet formed was resuspended in 200 µL of the Guava Muse Luminex^®^ Cell Cycle, and the tubes were incubated for 30 min at room temperature, away from light.

After incubation, the samples were read in Guava Muse Luminex^®^, and the data with the histograms generated, which contain information on the percentage of the different phases of the cell cycle, were analysed in the statistical program Prism6. This methodology was performed in duplicate of three independent experiments.

#### 3.6.4. Evaluation of PI3K Activation and Phosphorylation

The methodology evaluates the activation and phosphorylation of PI3K and was performed from the labelling with the Guava Muse Luminex^®^ PI3K Activation Dual Detection Kit (catalog n^o^. MCH200103, Luminex) containing the anti-AKT/PKB-PECy5 antibody to measure total PI3K and the anti-phospho-AKT (Ser473)-Alexa Fluor^®^ 555 antibody for measurement of activated PI3K.

Following 48 h of incubation in an incubator at 37 °C with 5% CO_2_, 1 mL of the cell solution was transferred to an Eppendorf tube, washed with PBS and centrifuged at 300× *g* for 5 min. The supernatant was discarded, and the pellet formed was resuspended in 500 µL of PBS. The samples were again centrifuged at 300× *g* for 5 min and the supernatant discarded.

Then the cell solutions were resuspended in 200 µL of assay buffer (Part no. CS202124), followed by the addition of 200 µL of fixation buffer (Part n^o^. CS202122), ending up with a total volume of 400 µL per sample. The samples were incubated for 5 min on ice and centrifuged at 300× *g* for 5 min and the supernatant discarded.

Cells were permeabilized by adding 200 µL of ice-cold permeabilization buffer (Part n^o^. CS202125) and then incubated on ice for 5 min and centrifuged at 300× *g* for 5 min, and the supernatant was discarded. The formed pellets were resuspended in 450 µL of assay buffer and vortexed at medium speed. In total, 90 µL of the solution from each sample was aliquoted into a new tube, and 10 µL of the cocktail solution containing Anti-phospho-Akt (Ser473), Alexa Fluor^®^555 and Anti-Akt/PKB, Pecy5 was added. The samples were homogenised and incubated for 30 min in the dark at room temperature.

After the incubation step, 100 µL of assay buffer was added per sample and centrifuged at 300× *g* for 5 min, and the supernatant was discarded. The formed pellets were resuspended in 200 µL of assay buffer and vortexed, and the samples were read in Guava Muse Luminex^®^. Data from three independent experiments performed in duplicate were analysed in the Prism6 program to determine the percentage of cells expressing activated and inactivated PI3K.

### 3.7. Microsomal Metabolic Stability

The metabolic stability of compound **5f** (LASSBio-2252) was studied as previously described [30], and 1 mg/mL of human rat microsomes was incubated with compound **2f** (10 μM) in the presence and in the absence of the cofactor (i.e., NADPH generating system). After mixing and incubation at 37 °C in a shaking water bath for 1 h, 500 µL cold acetonitrile containing the internal standard was added to terminate the reaction at different time intervals (0, 15, 30, 45 and 60 min). After mixing and centrifugation at 10,500× *g* for 15 min at 4 ºC, the supernatant was filtered and then was injected into the HPLC system using a mixture of methanol: water acidified (0.1% formic acid, pH 3.0) (50:50, *v/v*) as the mobile phase at a flow rate of 1.0 mL/min. After analysing the results, the pharmacokinetic parameters were calculated as microsomal half-life (t_1/2_) and apparent intrinsic clearance (Clapp) [31].

## 4. Conclusions

In summary, we describe the identification of a new cytotoxic agent, designed as a simplified analogue of gedatolisib. LASSBio-2252 (**5f**) was shown to have cytotoxic potency in the low micromolar range and with enhanced cytotoxic action on leukemic versus solid tumour cell lines. The cytotoxic effect was confirmed by different methodologies and in a comparative manner to gedatolisib. The ability of **5f** to exert its cytotoxic action through modulation of the PI3K pathway was demonstrated in a phenotypic model compared to gedatolisib. This new analogue has great aqueous solubility and high metabolic stability in rat liver microsomes. Taken together, LASSBio-2252 (**5f**) can be considered a new lead candidate, whose subsequent structural optimization steps are required to increase its cytotoxic potency.

## Figures and Tables

**Figure 1 pharmaceuticals-16-00209-f001:**
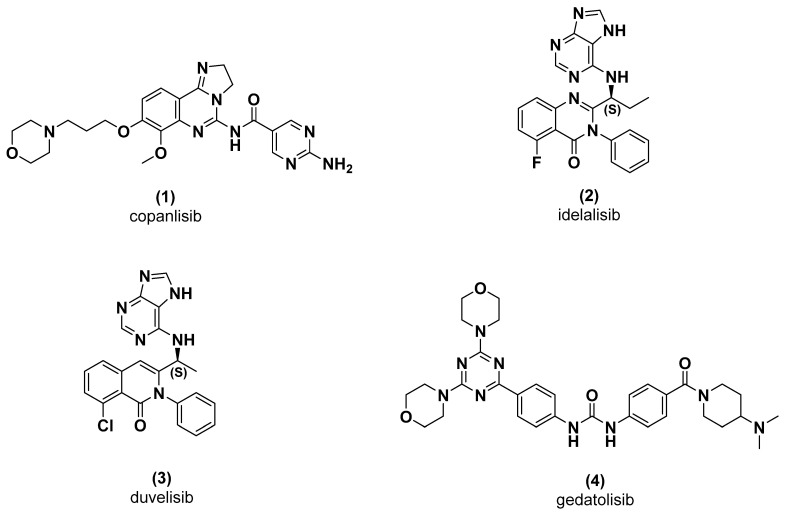
Examples of PI3K inhibitors.

**Figure 2 pharmaceuticals-16-00209-f002:**
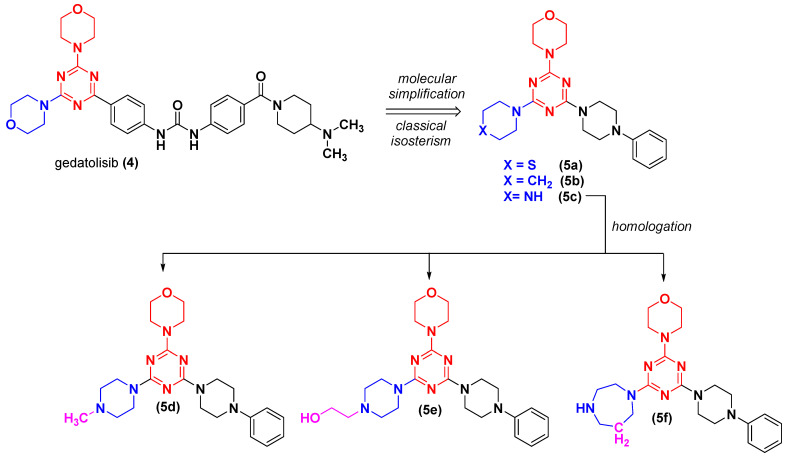
Design concept of compounds **5a-f** using gedatolisib as prototype.

**Figure 3 pharmaceuticals-16-00209-f003:**
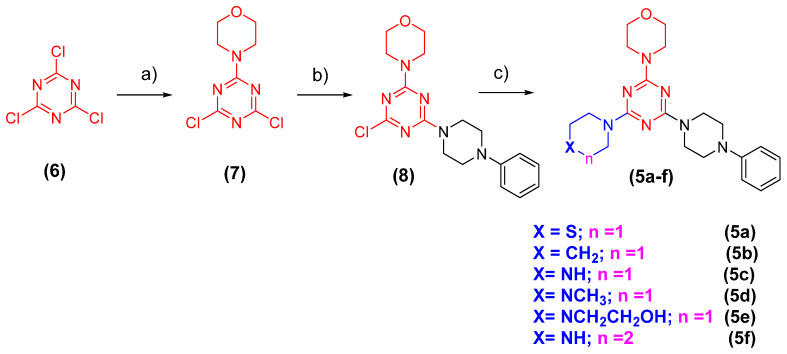
General methodology for synthesis of compounds **5a-f**. Reagents and conditions: (**a**) morpholine, Et_3_N, THF, −20 °C to 0 °C; 6 h; 91%; (**b**) phenylpiperazine, Et_3_N, THF, −20 °C to 0 °C; 24 h; 90%; (**c**) amines, Et_3_N, THF, reflux; 18–24 h; 50–58%.

**Figure 4 pharmaceuticals-16-00209-f004:**
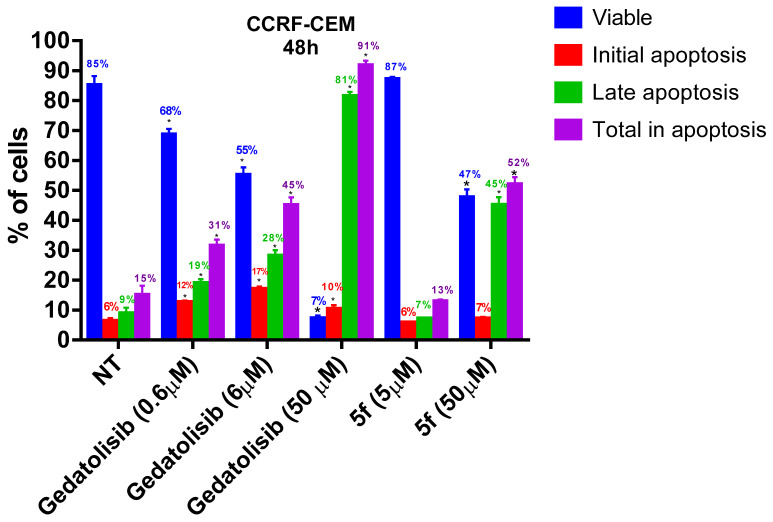
Evaluation of phosphatidylserine externalization and cell membrane integrity caused by gedatolisib (**4**) and LASSBio-2252 (**5f**) against CCRF-CEM cell line in 48 h. Data presented as mean ± standard error of the mean of three independent experiments. * *p* < 0.05 compared to the untreated group (DMSO 1%).

**Figure 5 pharmaceuticals-16-00209-f005:**
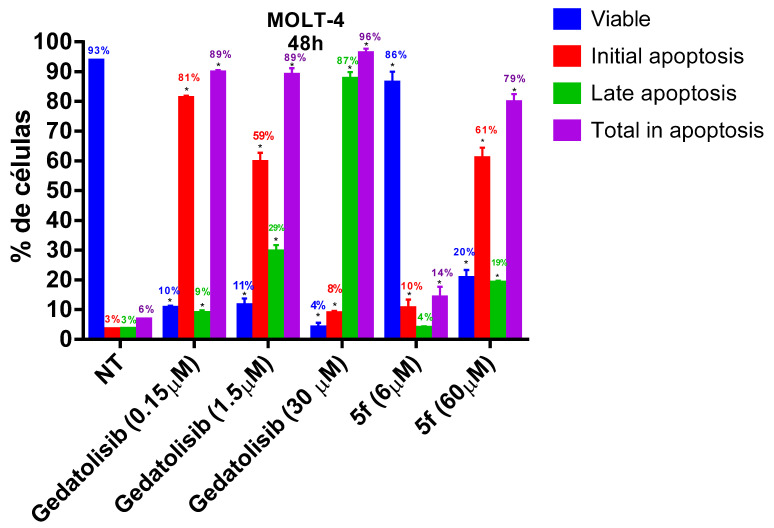
Evaluation of phosphatidylserine externalization and cell membrane integrity caused by gedatolisib (**4**) and LASSBio-2252 (**5f**) against MOLT-4 cell line in 48 h. Data presented as mean ± standard error of the mean of three independent experiments. * *p* < 0.05 compared to the untreated group (DMSO 1%).

**Figure 6 pharmaceuticals-16-00209-f006:**
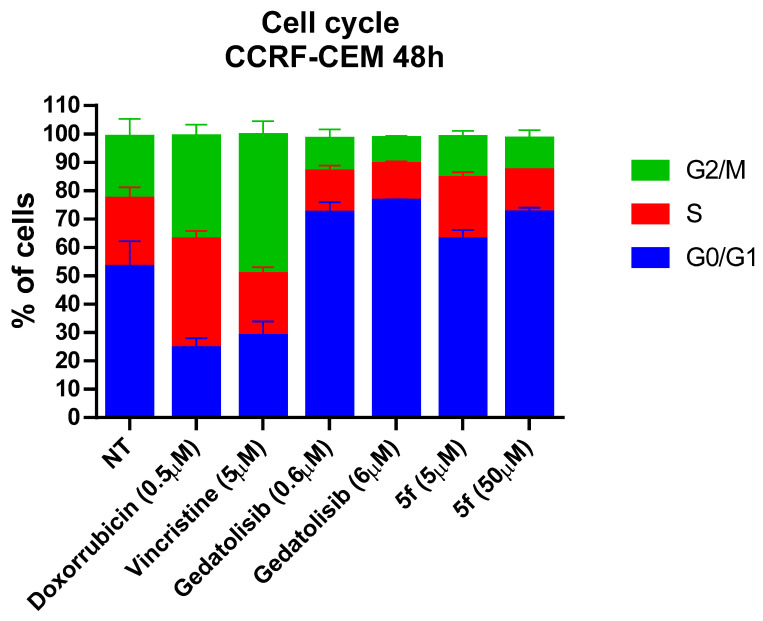
Cell cycle assessment by LASSBio-2252 (**5f**) and the standards doxorubicin, vincristine and gedatolisib on CCRF-CEM cell line in 48 h. Data presented as mean ± standard error of the mean of three independent experiments.

**Figure 7 pharmaceuticals-16-00209-f007:**
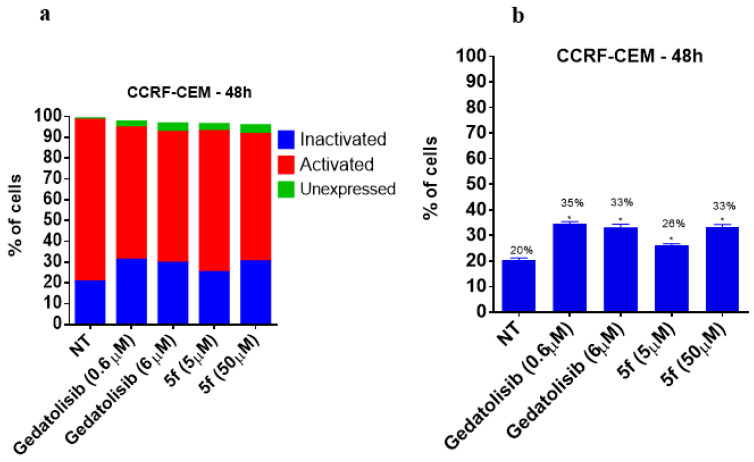
Effect of gedatolisib (**4**) and LASSBio-2252 (**5f**) on the PI3K activation and phosphorylation in CCRF-CEM cell line at 48 h. Data are presented as mean ± standard error of mean, * *p* < 0.05 compared to the untreated group (DMSO 1%); the graphs represent: (**a**) all PI3K tag states, (**b**) PI3K inactivated.

**Figure 8 pharmaceuticals-16-00209-f008:**
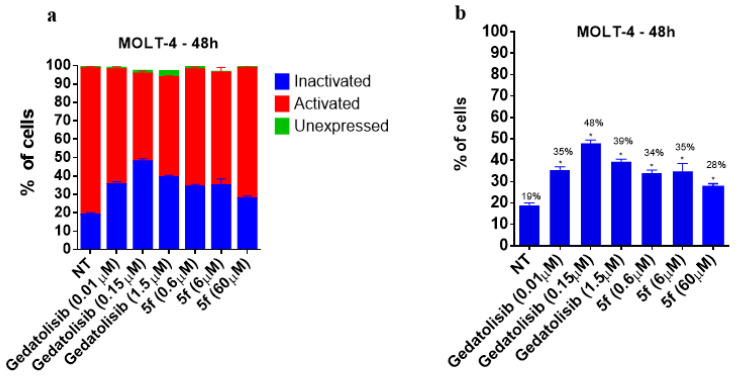
Effect of gedatolisib (**4**) and LASSBio-2252 (**5f**) on the PI3K activation and phosphorylation in MOLT-4 cell line at 48 h. Data are presented as mean ± standard error of mean, * *p* < 0.05 compared to the untreated group (DMSO 1%); the graphs represent: (**a**) all PI3K tag states, (**b**) PI3K inactivated.

**Table 1 pharmaceuticals-16-00209-t001:** pH-dependent aqueous solubility of compounds **5a-f** and their cLogP and cLogD values.

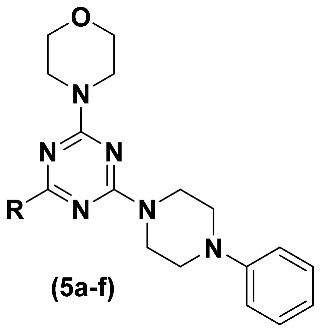						
**Compounds**	**R**	**Solubility pH 7.4** **(µM)**	**Solubility pH 5.8** **(µM)**	**cLogP**	**cLogD_7.4_**	**cLogD_5.8_**
**5a**	Thiomorpholine	2.88	0.52	3.42	2.65	0.84
**5b**	Piperidine	0.51	2.84	4.08	3.08	1.48
**5c**	Piperazine	59.13	43.35	2.58	1.08	−1.61
**5d**	*N*-methylpiperazine	50.92	30.69	2.97	1.81	−0.67
**5e**	Hydroxyethylpiperazine	49.15	31.19	2.7	1.84	−0.67
**5f**	Homopiperazine	42.38	39.33	2.96	0.09	1.25
**9** (CAS−942789959)	Morpholine	6.03	3.92	2.54	−1.77	−1.25

**Table 2 pharmaceuticals-16-00209-t002:** CC_50_ values calculated from the 72 h MTT cell viability assay.

Cell Line	Gedatolisib CC_50_ 72 h (µM)	GSK-1059615 CC_50_ (µM)
PC-3	0.54 (0.27–1.08) E_max_ = 80%	6.78 (3.46–13.26) E_max_: 67%
MCF7	0.07 (0.03–0.20) E_max_ = 87%	0.55 (0.27–1.12) E_max_: 80%
K562	24.09 (15.84–36.63) E_max_ = 64%	1.82 (1.24–2.68) E_max_ = 84%
CCRF-CEM	0.03 (0.01–0.05) E_max_ = 90%	0.43 (0.09–1.95) E_max_ = 93%
MOLT-4	0.02 (0.01–0.03) E_max_ = 93%	0.63 (0.13–3.07) E_max_ = 92%

Data expressed in micromolar (µM) and 95% confidence interval. Emax represents the maximum effect of the compound at its maximum concentration used. Emax values were reported when lower than 100%.

**Table 3 pharmaceuticals-16-00209-t003:** CC_50_ values calculated from the 72 h MTT cell viability assay.

Cell Line	5c (LASSBio-2247) CC_50_ (µM)	5d (LASSBio-2248) CC_50_ (µM)	5f (LASSBio-2252) CC_50_ (µM)
PC-3	92.29 (39.92–213.40) E_max_ = 58%	91.31 (60.23–138.40) Emax = 72%	62.15 (55.72–69.33) E_max_ = 84%
MCF7	>100 E_max_: 57%	>100 E_max_ = 77%	37.04 (31.82–43.12) E_max_ = 91%
K562	25.63 (18.83–33.13) E_max_ = 85%	50.22 (40.23–62.69) E_max_ = 80%	23.41 (16.31–33.59) E_max_ = 84%
CCRF-CEM	37.33 (28.61–48.71) E_max_ = 94%	66.50 (61.35–72.09) E_max_ = 95%	6.25 (4.10–9.54) E_max_ = 93%
MOLT-4	31.84 (23.48–43.17) E_max_ = 93%	64.07 (57.45–71.45) E_max_ = 91%	9.76 (5.44–17.53) E_max_ = 93%

Data expressed in micromolar (µM) and 95% confidence interval. Emax represents the maximum effect of the compound at its maximum concentration used. Emax values were reported when lower than 100%.

**Table 4 pharmaceuticals-16-00209-t004:** CC_50_ values calculated from the 48 h MTT cell viability assay.

Cell Line	Gedatolisib CC_50_ (µM)	5f (LASSBio-2252) CC_50_ (µM)
CCRF-CEM	0.14 (0.07–0.28) E_max_ = 94%	11.1 µM (5.53–22.29) E_max_ = 69%
MOLT-4	0.15 (0.06–0.31) E_max_ = 92%	36.48 (27.33–49.76) E_max_ = 80%

Data expressed in micromolar (µM) and 95% confidence interval. Emax represents the maximum effect of the compound at its maximum concentration used. Emax values were reported when lower than 100%.

**Table 5 pharmaceuticals-16-00209-t005:** CC_50_ values calculated using the Muse^®^ Count & Viability Kit on CCRF-CEM and MOLT-4 after 72 and 48 h of incubation.

Cell Line	Gedatolisib CC_50_ (µM) 72 h	Gedatolisib CC_50_ (µM) 48 h	5f CC_50_ (µM) 72 h	5f CC_50_ (µM) 48 h
CCRF-CEM	6.15 (4.60–8.24) Emax: 94%	5.88 (4.16–8.32) Emax: 94%	53.15 (50.30–56.17) Emax: 96%	48.96 (45.71–52.45) Emax: 95%
MOLT-4	1.15 (0.85–1.56) Emax: 86%	1.65 (1.24–2.20) Emax: 82%	60.78 (58.02–63.66) Emax: 92%	51.01 (47.89–58.67) Emax: 90%

Data expressed in micromolar (µM) and 95% confidence interval. Emax represents the maximum effect of the compound at its maximum concentration used. Emax values were reported when lower than 100%.

**Table 6 pharmaceuticals-16-00209-t006:** Microsomal metabolic stability of LASSBio-2252 (**5f**) in rat liver microsomes.

NADPH-Generating System	Metabolization Rates (%)	Elimination Rate Constant (k)	t_1/2_ (min)	Clapp (mL/min/g)	Recovery (%)
Present	2.90	0.0015	462.00	0.058	101,0903
Absent	9.09	0.0017	407.65	0.066	100,4186

Assay performed in triplicate; t_1/2_ (min) = 0.693/k; k: -(a) is the slope of the curve; Clapp (mL/min/mg) = (0.693/t_1/2_) × (incubation volume mL/microsomal protein mg) × (microsomal protein mg/liver weight g) × (liver weight g/rat body weight kg); incubation volume: 0.25 mL; microsomal protein: 1 mg; liver: 12.035 g; rat weight: 0.32025 kg.

## Data Availability

Data is contained within the article and Appendix A.

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
