# Peer review of "Design, Synthesis and Phenotypic Profiling of Simplified Gedatolisib Analogues"

_pharmaceuticals, 2023, doi:10.3390/ph16020209_

Round 1

Reviewer 1 Report

The authors described the identification of a simplified analogue of gedatolisib LASSBio-2252 (2f) which demonstrated low micromolar cellular inhibition potency leukemic cell lines. The authors also used multiple methods to evaluate the mechanism of 2f. Most importantly, 2f demonstrated promising drug-like properties which makes it a good candidate for future development. I would like to recommend accepting this paper for publication after the following issues are addressed.

1. The resolution for figure 1 and figure 2 is too low, please adjust.

2. I would suggest adding structures for typical PI3K inhibitors in the introduction part.

3. Did you run enzymatic assays for 2f, how is the selectivity?

Author Response

The authors would like to thank the referees and all the suggestions and corrections suggested.

Referee 1

The authors described the identification of a simplified analogue of gedatolisib LASSBio-2252 (2f) which demonstrated low micromolar cellular inhibition potency leukemic cell lines. The authors also used multiple methods to evaluate the mechanism of 2f. Most importantly, 2f demonstrated promising drug-like properties which makes it a good candidate for future development. I would like to recommend accepting this paper for publication after the following issues are addressed.

  1. The resolution for figure 1 and figure 2 is too low, please adjust.

RESPONSE: DONE

  1. I would suggest adding structures for typical PI3K inhibitors in the introduction part.

        RESPONSE: DONE

  1. Did you run enzymatic assays for 2f, how is the selectivity?

RESPONSE: Unfortunately, we did not. We do not have conditions to do that in our lab or to pay to do at Reaction Biology. These are the reasons that our studies were totally based in a phenotypic approach.

Reviewer 2 Report

Dear Author,

Your work under consideration is important and valuable work, but the most important shortcoming or organizational error that caught my attention is the rearrangement of Figure 1 and Figure 2. Because after Figure 1, compounds 3 and 4 are mentioned in the Material and method section. , but information about its structure is given in Figure 2 on the next page. Therefore, I advise you to disconnect this link. I also recommend using the more up-to-date bibliographies suggested in the study.

Finally, it would be good to review the results section in detail with more detailed causes and consequences.

Author Response

The authors would like to thank the referees and all the suggestions and corrections suggested.

Referee 2

Your work under consideration is important and valuable work, but the most important shortcoming or organizational error that caught my attention is the rearrangement of Figure 1 and Figure 2. Because after Figure 1, compounds 3 and 4 are mentioned in the Material and method section. , but information about its structure is given in Figure 2 on the next page. Therefore, I advise you to disconnect this link. I also recommend using the more up-to-date bibliographies suggested in the study.

Finally, it would be good to review the results section in detail with more detailed causes and consequences.

RESPONSE: DONE
